# Comparison of Discretionary Food and Drink Intake Based on a Short Web-Based Sugar-Rich Food Screener and a Validated Web-Based 7-Day Dietary Record

**DOI:** 10.3390/nu14061184

**Published:** 2022-03-11

**Authors:** Amanda Cramer-Nielsen, Sidse Marie Sidenius Bestle, Anja Pia Biltoft-Jensen, Jeppe Matthiessen, Anne Dahl Lassen, Bodil Just Christensen, Sarah Jegsmark Gibbons, Ellen Trolle

**Affiliations:** 1Division of Food Technology, National Food Institute, Technical University of Denmark, 2800 Kongens Lyngby, Denmark; amandac-n@outlook.com (A.C.-N.); simsib@food.dtu.dk (S.M.S.B.); jmat@food.dtu.dk (J.M.); adla@food.dtu.dk (A.D.L.); boju@food.dtu.dk (B.J.C.); s_jg_85@hotmail.com (S.J.G.); eltr@food.dtu.dk (E.T.); 2Independent Researcher, 2800 Kongens Lyngby, Denmark

**Keywords:** discretionary foods and drinks, dietary screener, sugar-rich food screener, web-based dietary assessment

## Abstract

A high consumption of discretionary foods and drinks has been associated with increased risk of multiple adverse health outcomes, including risk of overweight and dental caries. The family-based cluster randomized intervention study “Are you too sweet?” aimed at reducing the intake of discretionary foods and drinks in a population of children starting pre-school. As part of the intervention a new short web-based sugar-rich food screener (SRFS), was developed to make the parents and the school health nurses aware of the children’s intake of discretionary foods and drinks. In addition to the short assessment tool the parents also completed a validated web-based 7-day dietary record for the children. In the present study, estimates for intake of discretionary foods and drinks from the two assessment tools were compared (*n* = 80). There was significant correlation between estimates from the two assessment tools, but the SRFS provided lower estimates for intake of discretionary foods and drinks compared to the 7-day dietary record. The correlation coefficient between the two assessment tools was 0.49 (*p* < 0.001) and Kappa coefficient was 0.33. It is concluded that the SRFS can provide a fairly ranking of participants according to their intake of discretionary foods and drinks when compared to a validated 7-day dietary record. The screener may be a useful tool in practical settings, such as school health nurse consultations, in order to gain insight into the child’s sweet intake habits.

## 1. Introduction

High sugar intake from both foods and beverages has been associated with overweight in children and adults [1,2]. This association is even more apparent between high intake of sugar-sweetened beverages and increased body weight [3,4,5,6]. Previous studies have found a negative association between intake of added sugar and important micronutrients in the diet. This association is a result of foods with a high level of added sugar often being energy-dense, but low in micronutrients [7,8,9]. Furthermore, there is strong evidence that high intake of added sugar increases the risk of dental caries [2]. 

The foods that contribute the most to the intake of added sugar in the Danish population are candy, chocolate, and sugar-sweetened beverages [10]. Due to the possible health consequences of high sugar, and thereby risk of a high energy, intake, the World Health Organization (WHO), have suggested a maximum intake limit of 10 E% from free sugars in the diet and further health improvements by lowering the free sugar intake to less than 5 E% [2]. Similarly, The Nordic Nutrition Recommendations (NNR) advice to keep intake of added sugars below 10 E% [11]. 

New guidelines for maximum intake of energy dense, nutrient poor foods and drinks have been suggested for both children and adults [12]. The new maximum limits are developed for intake of energy dense, nutrient poor foods and drinks, also called discretionary foods and drinks. Discretionary foods and drinks include candy, chocolate, cookies, biscuits, desserts, cakes, snacks, and sugar-sweetened beverages. The maximum limits are based on a modeled diet, which meet the Danish Official Dietary Guidelines 2013 and Nordic Nutrition Recommendations 2012. The modeling of the diet is based on an average Danish diet from the Danish National Survey of Diet and Physical Activity 2011–2013. The maximum intake limit for discretionary foods and drinks is the difference between the energy intake in the modeled diet and the total energy needed for each age group [12]. These maximum intake limits for discretionary foods and drinks are now official Danish guidelines [13]. 

The Danish “Are you too sweet?” study was a multicomponent 3.5-month family-focused intervention study, which aimed at evaluating the effect of communicating the new guidelines on maximum intake of discretionary foods and drinks to families with children starting pre-school. In Denmark, all children starting pre-school are invited to a health consultation with a school health nurse. During the early phase of the “Are you too sweet” study, practitioners and especially school dentists were requesting information about the children’s dietary intake, to be able to target recommendations to them. It became clear that there was a need to develop an easy-to-use tool, where parents could register the child’s intake of discretionary foods and drinks, which could provide an overview of the child’s intake to both parents and practitioners. At the moment, food frequency questionnaires and food records are commonly used when assessing children’s food intake in research settings, but some of the assessment methods are both extensive and can be very time consuming [14].

In the “Are you too sweet” study, a web-based short assessment tool for estimating dietary intake of discretionary foods and drinks was developed. To our knowledge, this is the first web-based short assessment tool for assessing intake of discretionary foods and drinks in children, which also provides feedback directly to the user and to the health professional. There have been developed and validated short assessment tools for children for assessment of various food groups, including for instance fruit, vegetables, diary and discretionary foods, but to our knowledge not many are only focused on discretionary foods and drinks and the majority is based on FFQ answered on paper [14]. 

The aim of this study was to compare estimates of intake of discretionary foods and drinks measured by the newly developed sugar-rich food screener (SRFS) to estimates from a previously validated web-based 7-day dietary record in a secondary analysis. 

## 2. Materials and Methods

This is a secondary analysis of data from the “Are you too sweet?” study, where data from the intervention group are analyzed, as only this group of participants registered the intake of discretionary foods and drinks in both the 7-day dietary record and the SRFS. 

In short, the “Are you too sweet?” study invited pre-school starters (*n* = 439) in 2020 and their families at six schools in a municipality in Denmark (Hvidovre) to participate in the study. Eligible children are children starting public pre-school education at six selected schools in Hvidovre municipality in August 2020. There are no inclusion criteria regarding diet or disease. This municipality was chosen as the setting, because of the social diversity in the area, which represents the national mean for socio-economic status, education, and ethnicity. As the intervention is non-invasive, the local ethics committee confirmed that no official approval was needed (file no.: H-20036402). The study was conducted in accordance with the Declaration of Helsinki and written informed consent was obtained before participation.

At the beginning of the intervention study 160 families signed up. The details of the study have been described in the study protocol [15]. In total, 95 children, i.e., the intervention group of project “Are you to sweet” were included in this study. All study families completed a questionnaire and a 7-day dietary record for the child in the start of the intervention period. After the completion of the 7-day dietary record and shortly before the health consultation, the families were asked to complete a short web-based assessment tool (the sugar-rich food screener), to measure the intake of discretionary foods and drinks prior to the health consultation at the school.

### 2.1. Web-Based Dietary Assessment Methods

#### 2.1.1. Web-Based 7-Day Dietary Record

In the start and end of the study period the parents were asked to complete a web-based 7-day dietary record for their child. A first version of the web-based 7-day dietary record has previously been validated in a population of children aged 8–11 years in relation to intake of whole grain, n-3 polyunsaturated fatty acids, fruit, juice, and vegetables, as well as energy intake [16,17,18,19].

The parents were invited to a meeting, where they were introduced to the use of the 7-day dietary record by the researchers. All food and drinks the children consumed was registered in the record. The parents were told that they could either fill in the foods and drinks their child were consuming during the day or complete the entire dietary record in the evening after the last eating occasion. Each day was divided into three main meals—breakfast, lunch, and dinner—and three in-between meals. The food items were found in a list including over 1700 foods and beverages of which 217 were defined as discretionary foods and drinks. Here the parent could search for the specific foods that were consumed. The parents chose amounts of each food item from between 1–4 portion size images or wrote the actual portion size in grams or ml. When choosing a portion size image, the weight of that portion size would appear. There was a possibility of registering less or more than that portion by correcting the quantity if a proper portion size was not available in the images.

The 7-day dietary record also included questions about where and with whom the meals were eaten and if the child had eaten as usual during the day. If no candy or chocolate was registered, the parents were asked if any candy or chocolate might be forgotten in the previous registrations. If the parent responded yes to this question, they were redirected back and had another chance to register the missing food.

If the parents forgot to register the child’s food in the evening, they received a reminder email the following day. If they still had not registered the food intake, they received a text message on the phone on the second day after, to remind them to complete the dietary record.

#### 2.1.2. The Sugar-Rich Food Screener

The SRFS is a short web-based dietary assessment tool for estimating intake of discretionary foods and drinks. In the SRFS, parents were asked to register all discretionary foods and drinks their child had consumed during the last 7 days. The SRFS was developed for individual use and to provide direct feedback to the users and practitioners. The parents received an email with a link to the SRFS three days before their child’s scheduled health consultation. The link was sent approximately two weeks after they finished the dietary record, thus, one week between the two registration periods was most common. As the study is a part of a real-life intervention, practicalities made it impossible to harmonize the timespan between the registrations. If the parents had not completed the SRFS the day before the health consultation, a text message was sent to them as a reminder and one from the researcher team would call them, if they still had not completed the SRFS after receiving the text message.

First page of the SRFS was an introduction to the parents on how they should complete the tool and a specification of which foods and drinks to include in the registration. On the next page the parents were asked which days their child had been to school and if they attended any special activities during the week for instance parties, birthdays, play dates, or going to the movies. This was done to prompt the parents to remember any unusual eating patterns during the last week.

After this, the parents were shown different occasions on which their child might have consumed discretionary foods and drinks. For weekdays, the occasions include breakfast, packed lunch, special occasion or hand out snacks during school time (e.g., for birthdays), leisure activity, afternoon, and evening. For weekend-days, the occasions included breakfast, before noon, lunch, afternoon, and evening. The parents had to choose which discretionary foods and drinks the child had eaten at each occasion, if any, separately for weekdays and weekend-days. The parents selected all the eaten discretionary foods and drinks from a predefined list of 62 generic discretionary foods and drinks. The parent had to choose portion size from between 2–4 images of the specific itemfor each discretionary foods and drinks for each eating occasion., Screenshots from the SRFS can be seen in Appendix A Figure A1 and Figure A2. 

#### 2.1.3. Output from the Sugar-Rich Food Screener

When the SRFS was completed, a result page became visible to the parent. On this page the registered amounts of discretionary foods and drinks were shown in portion sizes equal to the portion size from the guidelines on maximum intake of discretionary foods and drinks developed at DTU.

On the feedback page the individual intake accumulated in portions per week was shown. The intake in portions was rounded to the nearest half portion, of discretionary foods and drinks the child had consumed the last week. The result was presented as an easy-to-follow pie chart, showing a green area with core foods and a red area showing the amount taken up by discretionary foods and drinks. The result page included both the individual results with the total number of portions the child had consumed and the recommended maximum intake of discretionary foods and drinks. A screen shot of the result page from the SRFS can be seen in Figure 1. The school health nurses received the information about the child’s intake of discretionary foods and drinks in text form. Beside including the total amount of portions, the information provided to the school health nurse, also included information about which type of food items was consumed, days and eating occasion of consumption, and if the child had any special activities during the week. They inserted the results in the child’s journal and used this information in the following health consultation. Based on the results from the SRFS, the school health nurses and families had detailed individual information on the child’s consumption of discretionary foods and drinks. This information provided a good starting point for discussing intakes and possible future changes. 

#### 2.1.4. Development of the Sugar-Rich Food Screener

The tool was developed in collaboration with a private company, which is specialized in serious games and e-learning. Health practitioners, researchers, and technical developers were all part of the development process from the beginning. Practitioners, both school health nurses and school dentists, provided their suggestions in order to develop the tool with focus on intended use in a practical setting. For instance, it was important that the result would be easy and quick to interpret from the child’s journal and that the feedback given to parents were objective and non-judging. From the research team’s point of view, it was important that data was reliable and used effective prompting in order to capture a week’s intake, while still being fast and easy to complete.

During the early phase of the “Are you too sweet?” study, a feasibility study was conducted with 18 parents participating. The researchers found that a version with non-chronological structure and more focus on meals and situations got better feedback and was evaluated, as being faster to complete. Therefore, it was chosen to use this structure in the SRFS.

The list of 62 discretionary foods and drinks in the SRFS was based on the list of discretionary foods and drinks from the 7-day dietary record. The food list in the 7-day dietary record included 1700 food and drinks, of which 217 items were discretionary food and drinks. The 217 discretionary food and drinks items were aggregated into categories to make 62 generic items for the SRFS. This aggregation of discretionary food and drink items were done by aggregating all similar food items into one food category. For instance, combining all different kind of chocolates bars, plain chocolate, filled chocolates and similar into one category called chocolate. 

Because this tool was developed to be used for registering children’s intake of discretionary foods and drinks, and to harmonize with the new guidelines for maximum intake of discretionary foods and drinks, smaller portion choices and new images were used for some food items in the SRFS. Furthermore, a normative conception of large portions for children was undesired in the SRFS. All images in the 7-day dietary record were manually screened with focus on any images that did not include suitable portions sizes for children. If none of the images illustrated children’s sized portions in an image series, new pictures were taken to provide appropriate options for the parents to choose from in the SRFS. Images of nine discretionary foods were determined being too large portion sizes or were not available in the existing images and new pictures were taken. The nine foods include rice biscuit with chocolate, pie/cake, ice pops, ice cream stick, cinnamon bun, layered cake, Danish pastry, chocolate biscuit, and dessert/fromage/pudding.

In the SRFS one portion was defined as 450 kJ. The recommended maximum limit for intake of discretionary foods and drinks for children aged 4–6 years is four portions per week. In this age group, one portion of discretionary foods can be replaced by intake of a sweetened beverage, which equals a portion of 250 mL [12]. The parents registered the amount of discretionary foods and drinks from pictures in the SRFS. These registrations were saved as amounts in g or mL depending on the food or drink chosen. The registrations in g and mL were converted into portions of 450 kJ.

### 2.2. Statistical Analysis

All statistical analysis was performed in R (CRAN, Version 3.6.3 (https://cran.r-project.org/ access date 10 January 2020)) and RStudio (PBC, Boston, MA, USA, Version 1.3.1073). 

The data analysis included all individuals who completed the SRFS and registered at least four days in the 7-day dietary record, where at least one of them was a weekend day. All intakes were converted to portions per week. When a child only had 4 to 6 registered days of intake in the 7-day food record, the intake was converted into a mean intake per day and multiplied by 7 days to estimate the amount per week. 

The primary analyses of the present study were the comparisons of the estimates of discretionary foods and drinks intake from the newly developed SRFS and the intake assessed by the previously validated web-based 7-day dietary record. 

The data from the 7-day dietary record was evaluated for both under- and over-reporters in relation to energy intake, which were calculated using the Goldberg cut-offs as suggested by Black and a PAL of 1.57 [20]. A visual inspection was conducted of the distribution to identify any outliers that might affect the outcomes of this study. Inclusion or exclusion of outliers was evaluated before further statistical analysis was conducted. Shapiro–Wilk test was performed on all data to examine the distribution of the sample and test for normal distribution. 

Means, medians, 25th and 75th percentiles for discretionary food and drink intake data from both assessment tools were evaluated to further describe the data. Wilcoxon signed rank test was conducted to test for any differences between the median intakes of discretionary foods and drinks reported in the two different assessment tools. 

To compare the intakes reported in the two different tools, correlation plots with intake of discretionary foods and drinks from both tools was made and Spearman’s correlation coefficient was calculated. 

To assess the agreement on an individual level, cross-classification and kappa statistics was investigated. The cross-classification was conducted on intake tertiles of discretionary foods and drinks (portions/week) estimated by the two methods. This was used to examine if the two methods classified participants in the same tertiles. Furthermore, Bland–Altman plots were made to visualize the accuracy of the intake estimates from the newly developed SRFS compared to the 7-day dietary record. These plots were interpreted in relation to the use in practical settings and any skewness to be aware of. 

Intake of discretionary foods and drinks were divided into food categories, to test if there were differences in the correlation between the two assessment tools, depending on the food category. The categories were ice cream and desserts; iced tea, cider and cordial, cakes and Danish pastries; biscuits and sweet snacks; candy and chocolate; salty snacks; soft drinks. The median of intake of foods in each food category reported in the two assessment tools, was calculated as portion and gram per week. Furthermore, Wilcoxon signed rank test and calculation of Spearman’s correlation coefficient were conducted to compare the intake of each food category. 

To explain any difference in intake of discretionary foods and drinks, the average registered portion size for each individual was compared for the two assessment tools. Furthermore, frequency of registrations for each child was compared for the two assessment tools. Frequency and number of parents forgetting to register any candy or chocolate in the 7-day dietary record was examined.

## 3. Results

Of the 95 participants included in this study 81 participants completed both the SRFS and the 7-day dietary record, in agreement with the description in the method section. One participant was removed from the sample due to an unusual high intake of discretionary drinks in both the SRFS and the 7-day dietary records. Thereby, 80 participants were included in the analyses. Of the 80 participants 73 children completed all 7 days of registration. Three children completed 5 days and 4 children completed 6 days.

All intake data from both dietary assessment tools were nonparametric.

In Table 1 is presented a summary of the characteristics of the study population. The majority of the children are of Danish ethnicity and classified as having a normal weight. The children were 5–7 years of age at baseline, with the majority of children being 6 years old.

All correlation coefficients for intake measured with the SRFS and the 7-day dietary record were significant, ranging from 0.43–0.45 for discretionary foods to 0.58–0.65 for discretionary drinks measured in g or ml per week and portions per week, respectively (*p* < 0.001). The same pattern was seen when looking at intake of discretionary foods and drinks together in portions per week, where the correlation coefficient was 0.49 (*p* < 0.001). A summary of the intake data from the SRFS and the 7-day dietary record can be seen in Table 2. The medians differ significantly between estimates of discretionary foods (*p* < 0.001) and drinks (*p* < 0.05) intake from the two dietary assessment tools, when running Wilcoxon signed rank test on the data. The SRFS is on average underestimating the discretionary food and drink intake by 27% when compared to the estimates from the 7-day dietary record. 

The relationship between the estimates of discretionary foods and drinks intake from the two assessment tools are illustrated in the Bland–Altman plots in Figure 2, Figure 3 and Figure 4. All Bland–Altman plots have wide limits of agreement and show a tendency towards the SRFS on average estimating lower intakes than the 7-day dietary record. Furthermore, the Bland–Altman plots indicate that there is a better agreement between the estimates from the two assessment tools for lower intakes compared to higher intakes for both discretionary foods and drinks.

In Table 3, the cross-classification of the intake of discretionary foods and drinks measured with the SRFS and the 7-day dietary record is illustrated. Forty participants were classified in the same tertile by the two dietary assessment tools and 72 participants were classified in the right or adjacent tertile by the two dietary assessment tools. The weighted kappa statistics is 0.33 (*p* < 0.001).

The analysis of intake data divided into discretionary food categories, as described in the method section, shows differences between the registered intake of discretionary foods and drinks from the SRFS and the 7-day dietary record. The intake of “Iced tea, cider, and cordials”, “cakes and Danish pastries”, “candy and chocolate”, “salty snacks”, and “soft drinks” all differs significantly between the SRFS and the 7-day dietary record. For the mentioned food categories, the intakes from the 7-day dietary record are higher than the intakes registered in the SRFS. There are significant correlation coefficients between the intakes registered in the two dietary assessment tools for “Iced tea, cider, and cordial” and “biscuits and sweet snacks”. The results can be seen in Table 4. 

There is a significant difference between the medians for the frequency of registered discretionary foods and drinks in the SRFS and the 7-day dietary record, but not any difference in the registered portion sizes for the two dietary assessment tools. The Spearman correlation coefficient between two assessment tools were 0.56 and 0.40 (*p* < 0.001) for the frequency and portion sizes, respectively. The medians and distributions of portion sizes and frequencies of registrations in the SRFS and the 7-day dietary record can be seen in Table 5 and Table 6.

Thirty-two percent of the participants registered, at least once, that they had forgotten to register candy or chocolate in the 7-day dietary record.

## 4. Discussion

There were significant correlation coefficients between the estimates from the SRFS and the 7-day dietary record ranging from 0.43–0.45 for discretionary foods to 0.58–0.65 for discretionary drinks. The cross-classification also indicates that the SRFS can rank participants into tertiles with fair agreement (Cohen’s kappa coefficient = 0.33) [21]. Even though the correlation coefficients are not very high, they are at a comparable level to what is found in other similar studies of nutritional research within the field [22,23]. The results suggests that the SRFS can be a useful tool to rank participants in relation to their intake levels of discretionary foods and drinks. However, the estimates for intake of discretionary foods and drinks from the SRFS were significantly lower than the estimates from the 7-day dietary record. This difference can both be a result of differences in the two tools but also because the two assessments were not conducted during the same time frame.

Similarly, Hendrie et al. found significant Pearson correlation coefficients of 0.44 for both beverages and “extra foods”, including energy-rich, nutrient poor foods, between a short food survey and 24-h dietary recall in children with a mean age of 7.1 years. The “extra foods” category included eight items and the beverage category included three items [22].

Another study by Bleiweiss–Sande et al. compared a new short FFQ with dietary intake assessed by a previously validated Block Kids Food Screener. The mean age of the children (*n* = 63) in the study was 9.9 years. They found Pearson correlation coefficients of 0.59 for salty snacks, 0.69 for sweet snacks, and 0.21 for sugar-sweetened beverages, but only the correlation coefficients for salty and sweet snacks were significant. In the study children registered their own food intake, which could have affected the correlations [23]. These results from the two studies are similar to the level of agreement seen in the present study.

Another study by Neuhouser et al. found Pearson correlation coefficient ranging from 0.66 to 0.71 for correlation between a food frequency questionnaire for beverages and “snacks and sweets” and intake assessed by 4-day dietary record. The sample (*n* = 46) had a mean age of 12.7 years. The validated FFQ included 19 items of which 17 were in either the beverage or “snacks and sweets” category. The study investigated intake during the last 7 days, but focused only on the frequency of consumption [24]. The correlation coefficients from this study are higher, but this could also be a result of the fact that they only investigated the frequency of registered intake and not amounts.

In the present study, the 7-day dietary record was completed each day, unlike the SRFS, where parents registered the intake for the last 7 days at once. This might have resulted in lower reported intake in the SRFS, compared to the 7-day dietary record due to recall bias. The analysis of the frequency of registrations in the two assessment tools, showed that the frequency was lower in the SRFS than the 7-day dietary record. This indicates that the parents forgot to register some of the discretionary food and drinks in the SRFS, which could be a result of the non-chronological approach in the SRFS. Furthermore, there is a difference in how the two assessment tools are prompting and guiding the parents through the dietary assessment. 

Some questions were added in the SRFS to try to minimize the possible underreporting. For instance, the parents were asked about their child’s attendance in any activities during the previous week. Besides being valuable information for the practitioners, this was done to help parents recall any occasions, where the child might have consumed more discretionary foods and drinks than during a usual day. Still this prompting might not have been sufficient to capture the entire intake as there is a significant difference between estimated intake of discretionary foods and drinks from the two assessments. 

As this is a secondary analysis, there are weaknesses in the design of the study, which could all have affected the results. One weakness of the study design is the fact that the registrations in the SRFS and the 7-day dietary record was completed in two different weeks. Ideally the study should have had a cross-over design. On group level however, this should not have affected the agreement, but it could have had an effect on the agreement on an individual level. Intake of discretionary foods and drinks may differ over time, among other reasons, due to differences in the activities attended each week. Conversely, if the participants had registered the intake in the same week, there would be a risk of the registration in the 7-day dietary record affecting the registration in the SRFS. This would both be due to an unusual focus on the child’s intake during the week, which could have made it easier to complete the SRFS. Another weakness of the study, caused by this being a secondary analysis, may be the relatively small number of participants. The sample size calculations were done with focus on the aim in the original “Are you too sweet” study and not the present study. However, all children completed both instruments and were within the same narrow age range.

The population of interest in this study were children starting pre-school. Due to low literacy in this age group, parents were instructed to complete the two dietary assessments. There is a risk of parents not being aware of all food and drinks their child consume during the day, when they are not together. 

Even though the estimates from the SRFS are lower than the estimates from the 7-day dietary record, it can be considered a useful tool, as the SRFS is able to rank the children in relation to their intake of discretionary foods and drinks. Furthermore, the SRFS should be both easier and faster for the families to complete. 

The SRFS was developed, with focus on it being a useful tool in a practical setting. With the information from the SRFS, the school health nurse can have an informed conversation with the family about the child’s intake habits and where any actions should be taken to change any unhealthy habits. The information from the SRFS would be just as relevant for the school dentists. They can often identify children with a high intake of discretionary foods and drinks due to dental caries. The school dentists would benefit from the short assessment tool, to be able to assess which areas to focus on with the parents and where to try to change the children’s habits regarding intake of discretionary foods and drinks.

This newly developed SRFS adds a new, easier, and faster way of estimating intake of discretionary foods and drinks. Even though the SRFS is a short assessment tool, it still provides information about the amount of specific foods eaten and not only frequency of consumption of food categories. 

## 5. Conclusions

This study showed that the intake of discretionary foods and drink estimated by the SRFS may provide a ranking of individuals in fair agreement with results from a validated 7-day dietary record. In addition, the SRFS have a tendency to underestimate the intake, however, bearing this in mind, the SRFS may be a useful dietary assessment tool to indicate the level of intake of discretionary foods and drinks on an individual level and useful in practical settings. It can provide information to practitioners about the individual child’s level of intake of discretionary food and drinks and if it is an important issue to address to improve the dietary habits and health of the child.

## Figures and Tables

**Figure 1 nutrients-14-01184-f001:**
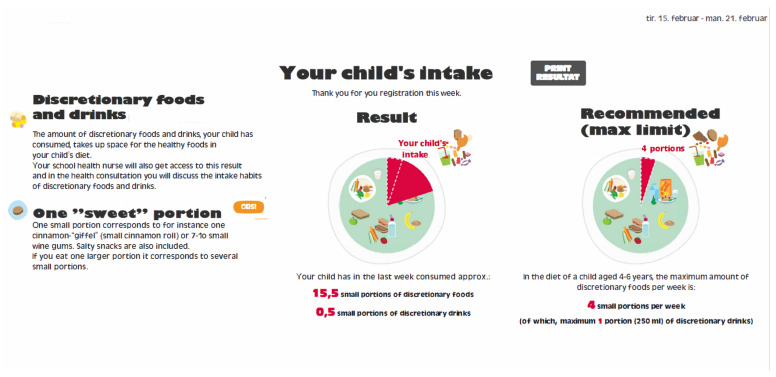
Screen shot of the result webpage from the SRFS. The pie chart to the right is showing the guideline for maximum intake of discretionary foods and drinks. The left pie chart shows the child’s actual intake of discretionary foods and drinks. To the left is explained what a portion of discretionary foods and drinks is and why these foods should be limited in a healthy diet. The original Danish text in this figure has been translated to English.

**Figure 2 nutrients-14-01184-f002:**
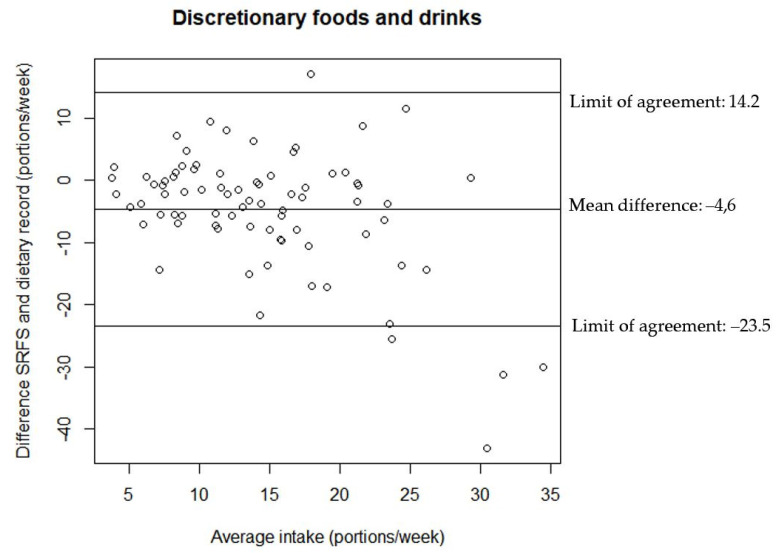
Agreement between the estimated portion intake of discretionary foods and drinks from the SRFS and the 7-day dietary record (*n* = 80).

**Figure 3 nutrients-14-01184-f003:**
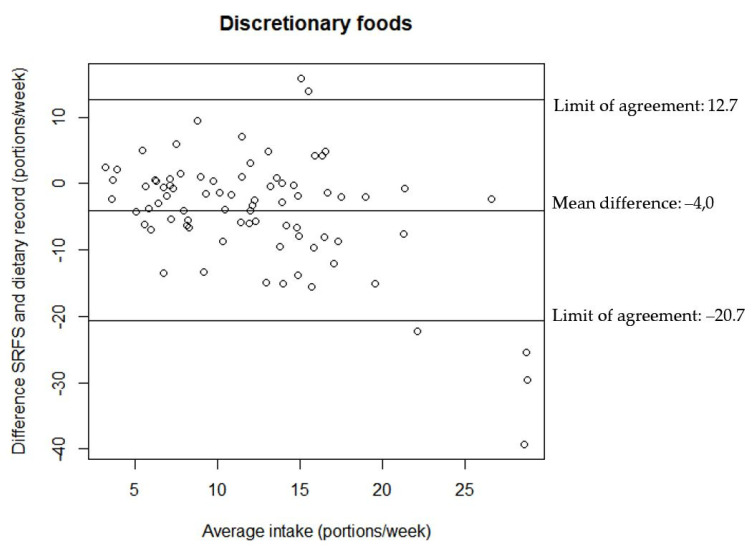
Agreement between the estimated portion intake of discretionary foods from the SRFS and the 7-day dietary record (*n* = 80).

**Figure 4 nutrients-14-01184-f004:**
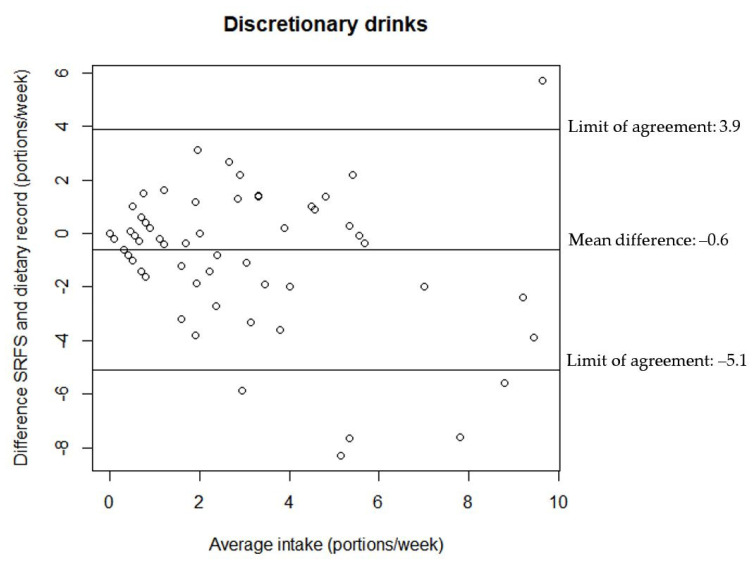
Agreement between the estimated portion intake of discretionary drinks from the SRFS and the 7-day dietary record (*n* = 80).

**Table 1 nutrients-14-01184-t001:** Summary of the characteristic of the study population.

	Population, *n* = 80
Sex; *n* (%)	
Girls	45 (56%)
Boys	35 (44%)
Age; mean (SD)	5.9 (0.3)
BMI; mean (SD)	15.7 (1.7)
Weight status; *n* (%)	
Underweight	1 (1%)
Normal weight	65 (81%)
Overweight	12 (15%)
Obese	1 (1%)
Highest parental education; *n* (%)	
Basic school (<12 years)	4 (5%)
Upper secondary school (12 years)	4 (5%)
Vocational education (13 y, practical)	16 (20%)
Short higher (13–14 years)	10 (13%)
Medium higher (15–16 years)	21 (26%)
Long higher (≥17 years)	25 (31%)
Ethnicity (maternal); *n* (%)	
Danish	73 (91%)
Other western	2 (3%)
Non-western	5 (6%)

**Table 2 nutrients-14-01184-t002:** Summary of intake of discretionary foods and drinks estimated by the SRFS and the 7-day dietary record in amount in portions of 450 kJ, g, or mL per week (*n* = 80).

	Sugar-Rich Food Screener					7-Day Dietary Record						
	Mean	95% CI	Median	25th Percentile	75th Percentile	Mean	95% CI	Median	25th Percentile	75th Percentile	Spearmans Correlation Coefficient	Wilcoxon Signed Rank Test
Portions of 450 kJ/week												
Total discretionary food and drink	12.2	[10.7; 13.6]	11	7.5	16.0	16.8	[14.6; 19.0]	14.8	9.5	21.0	0.49 *p* < 0.001	*p* < 0.001
Discretionary food	10.2	[9.1; 11.4]	9.8	6.4	13.5	14.3	[12.3; 16.2]	13.3	7.9	18.3	0.45 *p* < 0.001	*p* < 0.001
Discretionary drink	1.9	[1.4; 2.5]	1	0	3.1	2.5	[1.9; 3.2]	1.4	0.4	3.9	0.65 *p* < 0.001	*p* < 0.05
g/week												
Discretionary food	302	[264; 340]	288	190	361	416	[358; 473]	388	232	510	0.43 *p* < 0.001	*p* < 0.001
mL/week												
Discretionary drink	429	[301; 558]	200	0	600	635	[467; 803]	350	88	963	0.58 *p* < 0.001	*p*< 0.05

**Table 3 nutrients-14-01184-t003:** Cross-classification between intake of discretionary foods and drinks (portions of 450 kJ/week) estimated by the SRFS and the 7-day dietary record (*n* = 80).

	Sugar-Rich Food Screener				
**7-day dietary record**		1. tertile	2. tertile	3. tertile	Total
	1. tertile	16	7	4	27
	2. tertile	7	10	9	26
	3. tertile	4	9	14	27
	Total	27	26	27	80

**Table 4 nutrients-14-01184-t004:** Wilcoxon signed rank test and Spearman’s correlation coefficients for intake of discretionary foods and drinks (portions of 450 kJ/week) from the SRFS and the 7-day dietary record divided into categories (*n* = 80).

Discretionary Food and Drink (Portions/Week)	Median Intake Sugar-Rich Food Screener	Median Intake 7-Day Dietary Record	*p*-Value from Wilcoxon Signed Rank Test	Spearman’s Correlation Coefficient (*p*-Value)
Ice cream and desserts	1 (0; 1.9)	1.5 (0.4; 2.3)	*p* = 0.12	0.004 (*p* = 0.97)
Iced tea, cider and cordial	0.8 (0; 2)	1.4 (0.6; 2.8)	*p* < 0.05	0.39 (*p* < 0.005)
Cakes and Danish pastries	0 (0; 0.9)	3.7 (1.9; 7.1)	*p* < 0.001	−0.12 (*p* = 0.32)
Biscuits and sweet snacks	3 (1.1; 4.4)	2.6 (1.1; 4.0)	*p* = 0.96	0.41 (*p* < 0.005)
Candy and chocolate	0.1 (0; 0.8)	3.1 (2.1; 4.7)	*p* < 0.001	0 (*p* = 0.97)
Salty snacks Soft drinks	1.6 (0; 2.4) 0.4 (0; 1.4)	1.9 (0.7; 4.0) 1.7 (0.8; 3.7)	*p* < 0.05 *p* < 0.001	−0.17 (*p* = 0.17) 0.13 (*p* = 0.41)

**Table 5 nutrients-14-01184-t005:** Mean portion size of registered intakes of discretionary foods and drinks (portions of 450 kJ) in the SRFS and the 7-day dietary record (*n* = 80).

	Sugar-Rich Food Screener	7-Day Dietary Record	
	Average Amount Registered at Each Eating Occasion (Portions)	Average Amount Registered at Each Eating Occasion (Portions)	Wilcoxon Signed Rank Test
Mean	1.1	1.1	
95% CI	[1.0; 1.2]	[1.0; 1.2]	
Median	1.0	1.1	*p* = 0.49
25th percentile	0.8	0.8	
75th percentile	1.3	1.4	

**Table 6 nutrients-14-01184-t006:** Frequency of registrations of intake of discretionary foods and drinks in the SRFS and the 7-day dietary record (*n* = 80).

	Sugar-Rich Food Screener	7-Day Dietary Record	
	Number of Registrations of Intake	Number of Registrations of Intake	Wilcoxon Signed Rank Test
Mean	13.0	15.7	
95% CI	[11.3; 14.8]	[13.7; 17.6]	
Median	11.0	15.0	*p* < 0.01
25th percentile	8.0	8.0	
75th percentile	18.0	20.0	

## Data Availability

In accordance with Danish law, the confidential data used in this study can only be accessed through the servers at The Technical University of Denmark. Access is granted upon request if the applicant fulfils the criteria for access. DTU Food can be contacted by email: apbj@food.dtu.dk.

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
