# Peer review of "Comparison of Discretionary Food and Drink Intake Based on a Short Web-Based Sugar-Rich Food Screener and a Validated Web-Based 7-Day Dietary Record"

_nutrients, 2022, doi:10.3390/nu14061184_

Round 1

Reviewer 1 Report

This is an interesting study describing a well examined new tool. My main critiques are with the presentation. I feel the paper could use more overall focus/clarity and detail in the methods. Specific comments below.

-What was the rationale for limiting the sample to the intervention arm only?

-Seems to be a lot of information included in the methods that is not really relevant to the stated aim of the study which is to compare the new screener to 7-day food records. For example, 2.1.3. describing the output and how it was used in the intervention sounds great and like a useful awareness-building approach, but isn't really relevant to the stated aim of the study. Would focus more on describing the development and testing of the screener.

-Portions per week is used as the units, but some only had 4 days of data for their 7 day record. Was everyone normalized to the same time frame? If someone had only four days of data, how did you handle the missing days?

-For the section "2.1.4. Development of the sugar-rich food screener" please include all information related to how the screener was developed and only include information about how the screener was developed. For example, in an earlier section, you allude to the 62 screener items being derived form the 217 items appearing in the dietary records. I would include this information as part of the development of the screener and would provide some description of that process.

-Please clarify, you say the "target group for this tool was only children," this makes it sound like the children complete the tool.

-Could you provide more information on timing? I see at the end you discussed this being two different weeks, unless I missed it, I didn't not see much detail in the methods around timing and recall periods (e.g., same 7 days or different).

-Could use more detail in the analysis section. For example, it says "cross-classification and kappa statistics was investigated" and the next sentence mentions tertiles. Tertiles of what, specifically? I don't feel like there is enough detail given so that I could recreate the methods.

-Table 2 could be modified to allow the word "percentile" to fit on one line.

-The units are portions/week and ml, right? Would clarify throughout. For example, there are no units in table 4 except to say "(portions/week)" in the title, but weren't the drinks in ml? What does median intake of 1 refer to? Portions? But are drinks in portions or ml?

-Table 5 seems to show very similar portions sizes, but they are significantly different by the Wilcoxon signed rank test?

-What does this tools add that some of these others you mentioned in the discussion do not?

Reviewer 2 Report

Thank you very much for allowing me review the article entitle “Comparison of discretionary food and drink intake based on a short web-based sugar-rich food screener and a validated web-based 7-day dietary record” (nutrients-1578049) presented for the section “Nutrition Methodology & Assessment.

Based on the family-based cluster randomized intervention study “Are you too sweet?”. Being the study's aimed at reducing the intake of discretionary foods and drinks in a population of children starting pre-school. They compare estimates of intake of discretionary foods and drinks measured by the newly developed sugar-rich food screener (SRFS) to estimates from a previously validated web-based 7-day dietary record in a secondary analysis. (n=80).

Materials and methods

Information about the project “Are you too sweet?” It should be in the introduction, not in this section. In this section should be the methodology applied to this work.

Ethics committee approval should be described. What are the inclusion criteria for pre-school participants, but are the questionnaires validated in children between 8 and 11 years of age? How old are the children under study really? Who was invited to participate?

The procedure of the school nurses is well described.

Has a sample size calculation been performed for this study?

How many agreed to participate? How many completed the study? participation rate?

Figure 1 should be in English.

In the methodology I suggest making a Kappa Index.

Results

Table 1 is poor, it could be designed by age groups, since food consumption changes a lot. A test could also be added that would allow us to identify where there are statistically significant differences in the baseline characteristics of the population.

Table 2 and the graphs show the results of all the children.

Table 3 and 4 are very interesting

The discussion raises with these results suggests that the SRFS can be a useful tool to rank participants in relation to their intake levels of discretionary foods and drinks, but the correlation coefficient is only 0.43-0.45 and 0.58-0.65.

The conclusions given the heterogeneity of the children studied and the results should be re-written.

The objective and design are very interesting, but the sample studied is very heterogeneous and small.

Author Response

Thank you very much for your reviewing our manuscript and for your valuable comments.
We are convinced that your comments have improved the quality of the article.

Reviewer 2
Open Review

Comments and Suggestions for Authors

Thank you very much for allowing me review the article entitle “Comparison of discretionary food and drink intake based on a short web-based sugar-rich food screener and a validated web-based 7-day dietary record” (nutrients-1578049) presented for the section “Nutrition Methodology & Assessment.

Based on the family-based cluster randomized intervention study “Are you too sweet?”. Being the study's aimed at reducing the intake of discretionary foods and drinks in a population of children starting pre-school. They compare estimates of intake of discretionary foods and drinks measured by the newly developed sugar-rich food screener (SRFS) to estimates from a previously validated web-based 7-day dietary record in a secondary analysis. (n=80).

Materials and methods

Information about the project “Are you too sweet?” It should be in the introduction, not in this section. In this section should be the methodology applied to this work.

Thank you for this comment, we agree with this. We have modified section 2, Materials and Methods, were some text has been moved to the Introduction and other are deleted or rewritten. We have focused on only including information that is relevant for the present study.

Ethics committee approval should be described. What are the inclusion criteria for pre-school participants, but are the questionnaires validated in children between 8 and 11 years of age? How old are the children under study really? Who was invited to participate?

With regard to ethics committee approval this has been described in the protocol paper (Bestle et al., 2020):

Line 95-98: As the intervention is non-invasive, the local ethics committee confirmed that no official approval

is needed (file no.: H-20036402). The study is conducted in accordance with the Declaration of Helsinki and written informed consent of parents has been obtained before participation.

Line 91-93: Eligible children are children starting public pre-school education at the six selected schools in Hvidovre

municipality in August 2020. There is no inclusion criteria regarding diet or disease.  

And yes a first version of the web-based questionnaire was validated in children between 8-11 y, as mentioned line 128-131. The population under study here was a few years younger (5-7y), but as it is the parents who complete the questionnaire, it was considered to be a minor difference.

The procedure of the school nurses is well described.
Thank you for mentioning this.

Has a sample size calculation been performed for this study?

We have not conducted sample size calculations for the present study as this is a secondary analysis. Sample size calculations were made for the original “Are you to sweet?”-study, as described in the protocol paper, and no further calculations were performed:
Power calculations were performed based on 6–7 year-old children from the Danish National Survey of Diet and Physical Activity [28]. Calculations determined that 76 participants will be needed in both intervention and control groups to detect a 25% reduction in the intake of sugar-rich discretionary foods at a power of 80% and a 95% confidence interval. For a 25% reduction in intake of added sugar, 63 participants will be needed. To account for dropouts or loss of participants to follow-up, an enrolment of a minimum of 100 families in the intervention group and an additional 100 families in the control group was planned.” (Bestle et al., 2020)

How many agreed to participate? How many completed the study? participation rate?

160 children/families agreed to participate in the study and 95 was in the intervention group. 81 children/families completed both the SRFS and the 7-day dietary record in accordance with the requirements listed in the method section.

Line 315-319: In total 81 participants completed both the SRFS and the 7-day dietary record, in agreement with the description in the method section. Only 3 children completed 5 days of registration and 4 children complted 6 days. One participant was removed from the sample due to an unusual high intake of discretionary drinks in both the SRFS and the 7-day dietary record. This was done to avoid this outlier to affect the outcomes.

We have now described how many children was invited for the intervention study and how many agreed to participate – and how many was assigned to the intervention group. This has been added:

Line 89: In short, the “Are you too sweet?”-study invited pre-school starters (n=439)

Line 99: At the beginning of the intervention study 160 families signed up.

Line 121-123: In total 95 children i.e. the intervention group of project “Are you to sweet” were included in this study.

Line 315-316: In total 81 participants completed both the SRFS and the 7-day dietary record, in agreement with the description in the method section.

Figure 1 should be in English.

We have now translated the text in figure 1 to English and stated this in the figure title: The original Danish text in this figure has been translated to English.

In the methodology I suggest making a Kappa Index.

Thank you for pointing at this. We agree in making use of Kappa statistics. We actually did, and the coefficient or index is stated in line 359-363 We must agree that it has not been given the focus, which it should have. Now we have implemented this result in the discussion and the conclusion.

Line 401-404: The cross-classification also indicates that the SRFS can rank participants into tertiles with fair agreement (Cohen's kappa coefficient = 0.33). Even though the correlation coefficients are not very high, they are at a comparable level to what is found in other similar studies of nutritional research within the field

And in the conclusion

Line 485-491: The ability of the SRFS to rank individuals according to intake of discretionary food and drinks when compared to a validated 7-day dietary record was moderate. Although the SRFS maybe have the tendency to underestimate the intake, the tool may be a useful dietary assessment tool to indicate the level of intake of discretionary foods and drinks on an individual level and might be useful in practical settings

Results

Table 1 is poor, it could be designed by age groups, since food consumption changes a lot. A test could also be added that would allow us to identify where there are statistically significant differences in the baseline characteristics of the population.

We agree that the food consumption in different age groups changes a lot according to the age of children. In this study the children are 5-7 years of age (and only one child was 7 years). We have added the age range to make it clear, that the children are not very different in age (Line 323-324). After discussing this, we have come to the conclusion that the group of children are not that heterogenous, that we should divide the children by age. Comparison with data about the population of children in the communicipality of study would have been nice but such data for a direct comparison were not available.

Table 2 and the graphs show the results of all the children.

Yes, Table 2 and the graphs include all 80 children included in this study. We have added “n=80” to all tables and graphs to clarify this.

Table 3 and 4 are very interesting

The discussion raises with these results suggests that the SRFS can be a useful tool to rank participants in relation to their intake levels of discretionary foods and drinks, but the correlation coefficient is only 0.43-0.45 and 0.58-0.65.

Yes, it is correct that the correlation coefficients are not very high, but they are similar to what other studies have found within the nutritional field of research. As there is a variability of food consumption, the correlation coefficients will never be close to 1 within this field. Furthermore, the cross-classification suggests that the SRFS can classify the participants fairly in the same tertiles as the 7-day dietary record.

We have now added comments in the discussion:

Line 401-404:  The cross-classification also indicates that the SRFS can rank participants into tertiles with fair agreement (Cohen's kappa coefficient = 0.33). Even though the correlation coefficients are not very high, they are at a comparable level to what is found in other similar studies of nutritional research within the field

The conclusions given the heterogeneity of the children studied and the results should be re-written.

We agree, that the conclusion should be modified, and we have changed the wording, as mentioned above

Line 485-491: The ability of the SRFS to rank individuals according to intake of discretionary food and drinks when compared to a validated 7-day dietary record was moderate. Although the SRFS maybe have the tendency to underestimate the intake, the tool may be a useful dietary assessment tool to indicate the level of intake of discretionary foods and drinks on an individual level and might be useful in practical settings

The objective and design are very interesting, but the sample studied is very heterogeneous and small.

As mentioned above in this study the children are 5-7 years of age (and only one child was 7 years). We have added the age range to make it clear, that the children are not very different in age. However, we agree that the sample size need to be discussed and we have added this:

Line 458-462:  Another weakness of the study, caused by this being a secondary analysis, may be the relatively small number of participants. The sample size calculations were done with focus on the aim in the original “Are you too sweet”-study and not the present study. However, all children completed both instruments and were within the same narrow age range.

Round 2

Reviewer 2 Report

Thank you very much for allowing me to review, again, the article entitled “Comparison of discretionary food and drink intake based on a short web-based sugar-rich food screener and a validated web-based 7-day dietary record” (nutrients-1578049 ) presented for the section “Nutrition Methodology & Assessment.

In line 119 they indicate that the participants in the study are 95 children, this does not coincide with the summary that indicates that there are 80, I do not understand what 80 means since in material and methods it is indicated that the food items were found in a list including over 1700 foods and beverages of which was defined as discretionary foods and drinks, but line 177 says “The parents selected all the eaten discretionary foods and drinks from a predefined list of 62 foods and drinks”. Line 237 says “The list in the 7-day dietary record included 217 discretionary food and drink items”. while in results 81 participants are indicated, which are not 80 either as indicated in the summary. Table 3 refers to 80 participants again.

For all of which, please clarify what the study sample is and what items is being compared.

The conclusions do not fit the results obtained in the study.

Author Response

Nutrients

Manuscript: Comparison of Discretionary Food and Drink Intake Based on a Short Web-Based Sugar Rich Food Screener and a validated Web-Based 7-day Dietary Record.

Second round: Response to reviewer 2

Comments from reviewer 2 and reply

Thank you very much for allowing me to review, again, the article entitled “Comparison of discretionary food and drink intake based on a short web-based sugar-rich food screener and a validated web-based 7-day dietary record” (nutrients-1578049 ) presented for the section “Nutrition Methodology & Assessment.

We would like to thank the reviewer for reviewing our manuscript a second time. The feedback has helped us clarifying the number of participants, the aggregation of discretionary food and drinks from the 7-day food list to 62 generic discretionary food and drinks used in the Sugar-Rich Food Screener (SRFS), and to rewrite the conclusion. Please find our detailed response to the reviewer below. The reviewer’s comments are presented in normal font, while our responses follow in red.

In line 119 they indicate that the participants in the study are 95 children, this does not coincide with the summary that indicates that there are 80, I do not understand what 80 means since in material and methods it is indicated that the food items were found in a list including over 1700 foods and beverages of which was defined as discretionary foods and drinks, but line 177 says “The parents selected all the eaten discretionary foods and drinks from a predefined list of 62 foods and drinks”.

Below are the response to the question about participant number included.

Line 96–97 Methods-section: It is clarified that 95 children were included in this study.

Line  282-289 Results section: It is further clarified that out of the 95 participants included in this study, 81 participants completed both the SRFS and the 7-day dietary record, and that one participant were removed  due to unusual high intake of discretionary foods and drinks – leaving 80 participants to be included in the analysis.

Line 237 says “The list in the 7-day dietary record included 217 discretionary food and drink items”. while in results 81 participants are indicated, which are not 80 either as indicated in the summary. Table 3 refers to 80 participants again.

Below are the answer to the aggregation of discretionary food and drinks from the 7-day food list to 62 generic discretionary food and drinks used in the SRFS.

Line 207-212 Development of the sugar-rich food screener: It is now clarified that the food list in the SRFS included 62 discretionary food and drinks. These 62 items was based on the list of discretionary foods and drinks from the 7-day dietary record which included 1700 food and drinks in total. Of these 1700 food and drinks, 217 items were discretionary food and drinks (also stated in section 2.1.1 – web-based 7-day dietary record: Line 118). These 217 discretionary food items were aggregated into 62 generic foods to be included in the SRFS. There is already an example of this aggregation described in line 212-215 i.e. combining different kind of chocolates (bars, plain chocolate, filled chocolate etc.) from the 7-day diet record into one chocolate in the SRFS.

For all of which, please clarify what the study sample is and what items is being compared.

See the above clarification

The conclusions do not fit the results obtained in the study.

We agree that the conclusion needed improvements to fit the results better. We have rewritten the conclusion, to better reflect the uncertainties of the estimates from the SRFS compared with a total dietary registration (validated 7-day dietary record). In evaluating the kappa statistics we now refer to fair agreement according to Mary L. McHugh. Interrater reliability: the kappa statistic Biochem Med (Zagreb). 2012 Oct; 22(3): 276–282. https://www.ncbi.nlm.nih.gov/pmc/articles/PMC3900052/# (Line 370).

Line 453-456: This study showed that the intake of discretionary foods and drink estimated by the SRFS may provide a ranking of individuals in fair agreement with results from a validated 7-day dietary record. In addition, the SRFS have a tendency to underestimate the intake, however, bearing this in mind, the SRFS may be a useful dietary assessment tool to indicate the level of intake of discretionary foods and drinks on an individual level and useful in practical settings. It can provide information to practitioners about the individual child’s level of intake of discretionary food and drinks, and if it is an important issue to address to improve the dietary habits and health of the child. 
